# Quantifying how much sensory information in a neural code is relevant for behavior

**Giuseppe Pica**[1,2]
giuseppe.pica@iit.it

**Eugenio Piasini**[1]
eugenio.piasini@iit.it

**Houman Safaai**[1,3]
houman_safaai@hms.harvard.edu

**Caroline A. Runyan**[3,4]
runyan@pitt.edu

**Mathew E. Diamond**[5]
diamond@sissa.it

**Tommaso Fellin**[2,6]
tommaso.fellin@iit.it

**Christoph Kayser**[7,8]
christoph.kayser@uni-bielefeld.de

**Christopher D. Harvey**[3]
Christopher_Harvey@hms.harvard.edu

**Stefano Panzeri**[1,2]
stefano.panzeri@iit.it

[1] Neural Computation Laboratory, Center for Neuroscience and Cognitive Systems@UniTn,
Istituto Italiano di Tecnologia, Rovereto (TN) 38068, Italy
[2] Neural Coding Laboratory, Center for Neuroscience and Cognitive Systems@UniTn,
Istituto Italiano di Tecnologia, Rovereto (TN) 38068, Italy
[3] Department of Neurobiology, Harvard Medical School, Boston, MA 02115, USA
[4] Department of Neuroscience, University of Pittsburgh,
Center for the Neural Basis of Cognition, Pittsburgh, USA
[5] Tactile Perception and Learning Laboratory,
International School for Advanced Studies (SISSA), Trieste, Italy
[6] Optical Approaches to Brain Function Laboratory,
Istituto Italiano di Tecnologia, Genova 16163, Italy
[7] Institute of Neuroscience and Psychology, University of Glasgow, Glasgow, UK
[8] Department of Cognitive Neuroscience, Faculty of Biology,
Bielefeld University, Universitätsstr. 25, 33615 Bielefeld, Germany

## Abstract

Determining how much of the sensory information carried by a neural code contributes to behavioral performance is key to understand sensory function and neural information flow. However, there are as yet no analytical tools to compute this information that lies at the intersection between sensory coding and behavioral readout. Here we develop a novel measure, termed the information-theoretic intersection information $I_{\mathrm{II}}(S; R; C)$, that quantifies how much of the sensory information carried by a neural response $R$ is used for behavior during perceptual discrimination tasks. Building on the Partial Information Decomposition framework, we define $I_{\mathrm{II}}(S; R; C)$ as the part of the mutual information between the stimulus $S$ and the response $R$ that also informs the consequent behavioral choice $C$. We compute $I_{\mathrm{II}}(S; R; C)$ in the analysis of two experimental cortical datasets, to show how this measure can be used to compare quantitatively the contributions of spike timing and spike rates to task performance, and to identify brain areas or neural populations that specifically transform sensory information into choice.

# 1   Introduction

Perceptual discrimination is a brain computation that is key to survival, and that requires both encoding accurately sensory stimuli and generating appropriate behavioral choices (Fig.1). Previous studies have mostly focused separately either on the former stage, called *sensory coding*, by analyzing how neural activity encodes information about the external stimuli [1, 2, 3, 4, 5, 6, 7, 8, 9, 10], or on the latter stage, called *behavioral readout*, by analyzing the relationships between neural activity and choices in the absence of sensory signal or at fixed sensory stimulus (to eliminate spurious choice variations of neural response due to stimulus-related selectivity) [11, 12, 13]. The separation between studies of sensory coding and readout has led to a lack of consensus on what is the neural code, which here we take as the key set of neural activity features for perceptual discrimination. Most studies have in fact defined the neural code as the set of features carrying the most sensory information [1, 2, 8], but this focus has left unclear whether the brain uses the information in such features to perform perception [14, 15, 16].

Recently, Ref. [17] proposed to determine if neural sensory representations are behaviorally relevant by evaluating the association, in single trials, between the information about the sensory stimuli $S$ carried by the neural activity $R$ and the behavioral choices $C$ performed by the animal, or, in other words, to evaluate the *intersection between sensory coding and behavioral readout*. More precisely, Ref. [17] suggested that the hallmark of a neural feature $R$ being relevant for perceptual discrimination is that the subject will perform correctly more often when the neural feature $R$ provides accurate sensory information. Ref.[17] proposed to quantify this intuition by first decoding sensory stimuli from single-trial neural responses and then computing the increase in behavioral performance when such decoding is correct. This intersection framework provides several advantages with respect to earlier approaches based on computing the correlations between trial-averaged psychometric performance and trial-averaged neurometric performance [13, 14, 18], because it quantifies associations between sensory information coding and choices within the same trial, instead of considering the similarity of trial-averaged neural stimulus coding and trial-averaged behavioral performance. However, the intersection information measure proposed in Ref.[17] relies strongly on the specific choice of a stimulus decoding algorithm, that might not match the unknown decoding algorithms of the brain. Further, decoding only the most likely stimulus from neural responses throws away part of the full structure in the measured statistical relationships between $S$, $R$ and $C$ [3].

To overcome these limitations, here we convert the conceptual notions described in [17] into a novel and rigorous definition of *information-theoretic intersection information between sensory coding and behavioral readout* $I_{\mathrm{II}}(S; R; C)$. We construct the information-theoretic intersection $I_{\mathrm{II}}(S; R; C)$ by building on recent extensions of classical information theory, called Partial Information Decompositions (PID), that are suited to the analysis of trivariate systems [19, 20, 21]. We show that $I_{\mathrm{II}}(S; R; C)$ is endowed with a set of formal properties that a measure of intersection information should satisfy. Finally, we use $I_{\mathrm{II}}(S; R; C)$ to analyze both simulated and real cortical activity. These applications show how $I_{\mathrm{II}}(S; R; C)$ can be used to quantitatively redefine the neural code as the set of neural features that carry sensory information which is also used for task performance, and to identify brain areas where sensory information is read out for behavior.

# 2   An information-theoretic definition of intersection information

Throughout this paper, we assume that we are analyzing neural activity recorded during a perceptual discrimination task (Fig.1). Over the course of an experimental trial, a stimulus $s \in \{s_1, ..., s_{N_s}\}$ is presented to the animal while simultaneously some neural features $r$ (we assume that $r$ either takes discrete values or is discretized into a certain number of bins) and the behavioral choice $c \in \{c_1, ..., c_{N_c}\}$ are recorded. We assume that the joint probability distribution $p(s, r, c)$ has been empirically estimated by sampling these variables simultaneously over repeated trials. After the animal learns to perform the task, there will be a statistical association between the presented stimulus $S$ and the behavioral choice $C$, and the Shannon information $I(S : C)$ between stimulus and choice will therefore be positive.

How do we quantify the intersection information between the sensory coding $s \rightarrow r$ and the consequent behavioral readout $r \rightarrow c$ that involves the recorded neural activity features $r$ in the same trial? Clearly, the concept of intersection information must require the analysis of the full trivariate probability distribution $p(s, r, c)$ during perceptual discriminations. The well-established,

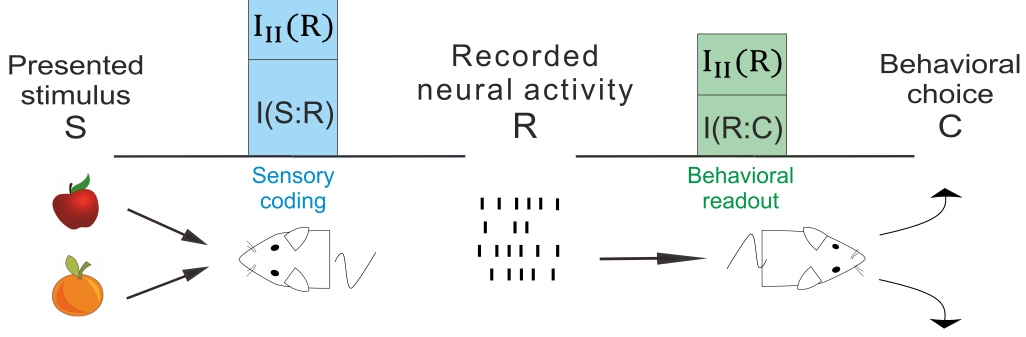

Figure 1: Schematics of the information flow in a perceptual discrimination task: sensory information $I(S : R)$ (light blue block) is encoded in the neural activity $R$. This activity informs the behavioral choice $C$ and so carries information about it ($I(R : C)$, green block). $I_{II}(S; R; C)$ is both a part of $I(S : R)$ and of $I(R : C)$, and corresponds to the sensory information used for behavior.

classical tools of information theory [22] provide a framework for assessing statistical associations between two variables only. Indeed, Shannon's mutual information allows us to quantify the sensory information $I(S : R)$ that the recorded neural features carry about the presented stimuli [3] and, separately, the choice information $I(R : C)$ that the recorded neural features carry about the behavior. To assess intersection information in single trials, we need to extend the classic information-theoretic tools to the trivariate analysis of $S, R, C$.

More specifically, we argue that an information-theoretic measure of intersection information should quantify *the part of the sensory information which also informs the choice*. To quantify this concept, we start from the tools of the Partial Information Decomposition (PID) framework. This framework decomposes the mutual information that two stochastic variables (the sources) carry about a third variable (the target) into four nonnegative information components. These components characterize distinct information sharing modes among the sources and the target on a finer scale than Shannon information quantities [19, 20, 23, 24].

In our analysis of the statistical dependencies of $S, R, C$, we start from the mutual information $I(C : (S, R))$ that $S$ and $R$ carry about $C$. Direct application of the PID framework then leads to the following nonnegative decomposition:

$$I(C : (S, R)) = SI(C : \{S; R\}) + CI(C : \{S; R\}) + UI(C : \{S \setminus R\}) + UI(C : \{R \setminus S\}), \quad (1)$$

where $SI$, $CI$ and $UI$ are respectively *shared* (or *redundant*), *complementary* (or *synergistic*) and *unique* information quantities as defined in [20]. More in detail,

- $SI(C : \{S; R\})$ is the information about the choice that we can extract from *any* of $S$ and $R$, i.e. the redundant information about $C$ shared between $S$ and $R$.

- $UI(C : \{S \setminus R\})$ is the information about the choice that we can only extract from the stimulus but not from the recorded neural response. It thus includes stimulus information relevant to the behavioral choice that is not represented in $R$.

- $UI(C : \{R \setminus S\})$ is the information about the choice that we can only extract from the neural response but not from the stimulus. It thus includes choice information in $R$ that arises from stimulus-independent variables, such as level of attention or behavioral bias.

- $CI(C : \{S; R\})$ is the information about choice that can be only gathered if both $S$ and $R$ are simultaneously observed with $C$, but that is not available when only one between $S$ and $R$ is simultaneously observed with $C$. More precisely, it is that part of $I(C : (S, R))$ which does not overlap with $I(S : C)$ nor with $I(R : C)$ [19].

Several mathematical definitions for the PID terms described above have been proposed in the literature [19, 20, 23, 24]. In this paper, we employ that of Bertschinger *et al.* [20], which is widely used for tripartite systems [25, 26]. Accordingly, we consider the space $\Delta_p$ of all probability distributions $q(s, r, c)$ with the same pairwise marginal distributions $q(s, c) = p(s, c)$ and $q(r, c) =$

$p(r, c)$ as the original distribution $p(s, r, c)$. The redundant information $SI(C : \{S; R\})$ is then defined as the solution of the following convex optimization problem on the space $\Delta_p$ [20]:

$$SI(C : \{S; R\}) \equiv \max_{q \in \Delta_p} CoI_q(S; R; C), \tag{2}$$

where $CoI_q(S; R; C) \equiv I_q(S : R) - I_q(S : R|C)$ is the co-information corresponding to the probability distribution $q(s, r, c)$. All other PID terms are then directly determined by the value of $SI(C : \{S; R\})$[19].

However, none of the existing PID information components described above fits yet the notion of intersection information, as none of them quantifies *the part of sensory information $I(S : R)$ carried by neural activity $R$ that also informs the choice $C$*. The PID quantity that seems to be closest to this notion is the redundant information that $S$ and $R$ share about $C$, $SI(C : \{S; R\})$. However, previous works pointed out the subtle possibility that even two statistically independent variables (here, $S$ and $R$) can share information about a third variable (here, $C$) [23, 27]. This possibility rules out using $SI(C : \{S; R\})$ as a measure of intersection information, since we expect that a neural response $R$ which does not encode stimulus information (i.e., such that $S \perp\!\!\!\perp R$) cannot carry intersection information.

We thus reason that the notion of intersection information should be quantified as the part of the redundant information that $S$ and $R$ share about $C$ that is also a part of the sensory information $I(S : R)$. This kind of information is even finer than the existing information components of the PID framework described above, and we recently found that comparing information components of the three different Partial Information Decompositions of the same probability distribution $p(s, r, c)$ leads to the identification of finer information quantities [21]. We take advantage of this insight to quantify the intersection information by introducing the following new definition:

$$I_{\mathrm{II}}(S; R; C) = \min\{SI(C : \{S; R\}), SI(S : \{R; C\})\}. \tag{3}$$

This definition allows us to further decompose the redundancy $SI(C : \{S; R\})$ into two nonnegative information components, as

$$SI(C : \{S; R\}) = I_{\mathrm{II}}(S; R; C) + X(R), \tag{4}$$

where $X(R) \equiv SI(C : \{S; R\}) - I_{\mathrm{II}}(S; R; C) \geq 0$. This finer decomposition is useful because, unlike $SI(C : \{S; R\})$, $I_{\mathrm{II}}(S; R; C)$ has the property that $S \perp\!\!\!\perp R \implies I_{\mathrm{II}}(S; R; C) = 0$ (see Supp. Info Sec.1). This is a first basic property that we expect from a meaningful definition of intersection information. Moreover, $I_{\mathrm{II}}(S; R; C)$ satisfies a number of additional important properties (see proofs in Supp. Info Sec. 1) that a measure of intersection information should satisfy:

1. $I_{\mathrm{II}}(S; R; C) \leq I(S : R)$: intersection information should be a part of the sensory information extractable from the recorded response $R$ – namely, the part which is relevant for the choice;

2. $I_{\mathrm{II}}(S; R; C) \leq I(R : C)$: intersection information should be a part of the choice information extractable from the recorded response $R$ – namely, the part which is related to the stimulus;

3. $I_{\mathrm{II}}(S; R; C) \leq I(S : C)$: intersection information should be a part of the information between stimulus and choice – namely, the part which can be extracted from $R$;

4. $I_{\mathrm{II}}(S; \{R_1, R_2\}; C) \geq I_{\mathrm{II}}(S; R_1; C), I_{\mathrm{II}}(S; R_2; C)$, as the task-relevant information that can be extracted from any recorded neural features should not be smaller than the task-relevant information that can be extracted from any subset of those features.

The measure $I_{\mathrm{II}}(S; R; C)$ thus translates all the conceptual features of intersection information into a well-defined analytical tool: Eq.3 defines how $I_{\mathrm{II}}(S; R; C)$ can be computed numerically from real data once the distribution $p(s, r, c)$ is estimated empirically. In practice, the estimated $p(s, r, c)$ defines the space $\Delta_p$ where the problem defined in Eq.2 should be solved. We developed a gradient-descent optimization algorithm to solve these problems numerically with a Matlab package that is freely available for download and reuse through Zenodo and Github https://doi.org/10.5281/zenodo.850362 (see Supp. Info Sec. 2). Computing $I_{\mathrm{II}}(S; R; C)$ allows the experimenter to estimate that portion of the sensory information in a neural code $R$ that is read out for behaviour during a perceptual discrimination task, and thus to quantitatively evaluate hypotheses about neural coding from empirical data.

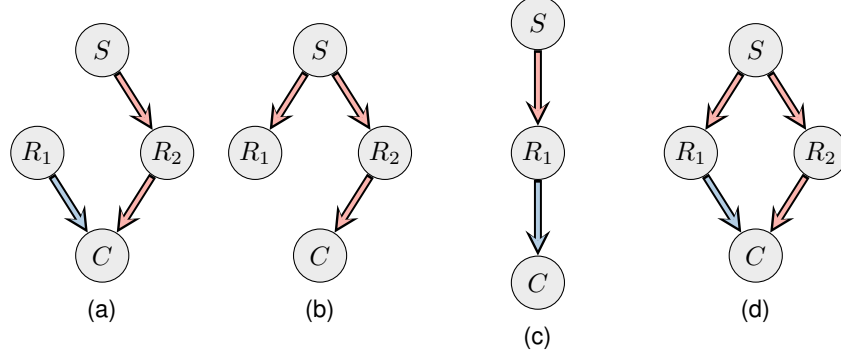

Figure 2: Some example cases where $I_{\mathrm{II}}(S; R_1; C) = 0$ for a neural code $R_1$. Each panel contains a probabilistic graphical model representation of $p(s, r, c)$, augmented by a color code illustrating the nature of the information carried by statistical relationships between variables. Red: information about the stimulus; blue: information about anything else (internal noise, distractors, and so on). $I_{\mathrm{II}}(R_i) > 0$ only if the arrows linking $R_i$ with $S$ and $C$ have the same color. a: $I(S : R_2) > I(S : R_1) = 0$. $I(C : R_2) = I(C : R_1)$. $I_{\mathrm{II}}(R_2) > I_{\mathrm{II}}(S; R_1; C) = 0$. b: $I(S : R_2) = I(S : R_1)$. $I(C : R_2) > I(C : R_1) = 0$. $I_{\mathrm{II}}(R_2) > I_{\mathrm{II}}(S; R_1; C) = 0$. c: $I(S : R_1) > 0$, $I(C : R_1) > 0$, $I(S : C) = 0$. d: $I(S : R_1) > 0$, $I(C : R_1) > 0$, $I(S : C) > 0$, $I_{\mathrm{II}}(S; R_1; C) = 0$.

## 2.1 Ruling out neural codes for task performance

A first important use of $I_{\mathrm{II}}(S; R; C)$ is that it permits to rule out recorded neural features as candidate neural codes. In fact, the neural features $R$ for which $I_{\mathrm{II}}(S; R; C) = 0$ cannot contribute to task performance. It is interesting, both conceptually and to interpret empirical results, to characterize some scenarios where $I_{\mathrm{II}}(S; R_1; C) = 0$ for a recorded neural feature $R_1$. $I_{\mathrm{II}}(S; R_1; C) = 0$ may correspond, among others, to one of the four scenarios illustrated in Fig.2:

- $R_1$ drives behavior but it is not informative about the stimulus, i.e. $I(R_1 : S) = 0$ (Fig.2a);

- $R_1$ encodes information about $S$ but it does not influence behavior, i.e. $I(R_1 : C) = 0$ (Fig.2b);

- $R_1$ is informative about both $S$ and $C$ but $I(S : C) = 0$ (Fig.2c, see also Supp. Info Sec.2);

- $I(S : R_1) > 0$, $I(R_1 : C) > 0$, $I(S : C) > 0$, but the sensory information $I(S : R_1)$ is not read out to drive the stimulus-relevant behavior and, at the same time, the way $R_1$ affects the behaviour is not related to the stimulus (Fig.2d, see also Supp. Info Sec.2).

## 3 Testing our measure of intersection information with simulated data

To better illustrate the properties of our measure of information-theoretic intersection information $I_{\mathrm{II}}(S; R; C)$, we simulated a very simple neural scheme that may underlie a perceptual discrimination task. As illustrated in Fig.3a, in every simulated trial we randomly drew a stimulus $s \in \{s_1, s_2\}$ which was then linearly converted to a continuous variable that represents the neural activity in the simulated sensory cortex. This stimulus-response conversion was affected by an additive Gaussian noise term (which we term "sensory noise") whose amplitude was varied parametrically by changing the value of its standard deviation $\sigma_S$. The simulated sensory-cortex activity was then separately converted, with two distinct linear transformations, to two continuous variables that simulated two higher-level brain regions. These two variables are termed "parietal cortex" ($R$) and "bypass pathway" ($R'$), respectively. We then combined $R$ and $R'$ with parametrically tunable weights (we indicate the ratio between the $R$-weight and the $R'$-weight with $\alpha$, see Supp. Info Sec.4) and added Gaussian noise (termed "choice noise"), whose standard deviation $\sigma_C$ was varied parametrically, to eventually produce another continuous variable that was fed to a linear discriminant. We took as the simulated behavioral choice the binary output of this final linear discriminant, which in our model was meant to represent the readout mechanism in high-level brain regions that inform the motor output.

We ran simulations of this model by varying parametrically the sensory noise $\sigma_S$, the choice noise $\sigma_C$, and the parietal to bypass ratio $\alpha$, to investigate how $I_{\mathrm{II}}(S; R; C)$ depended on these parameters.

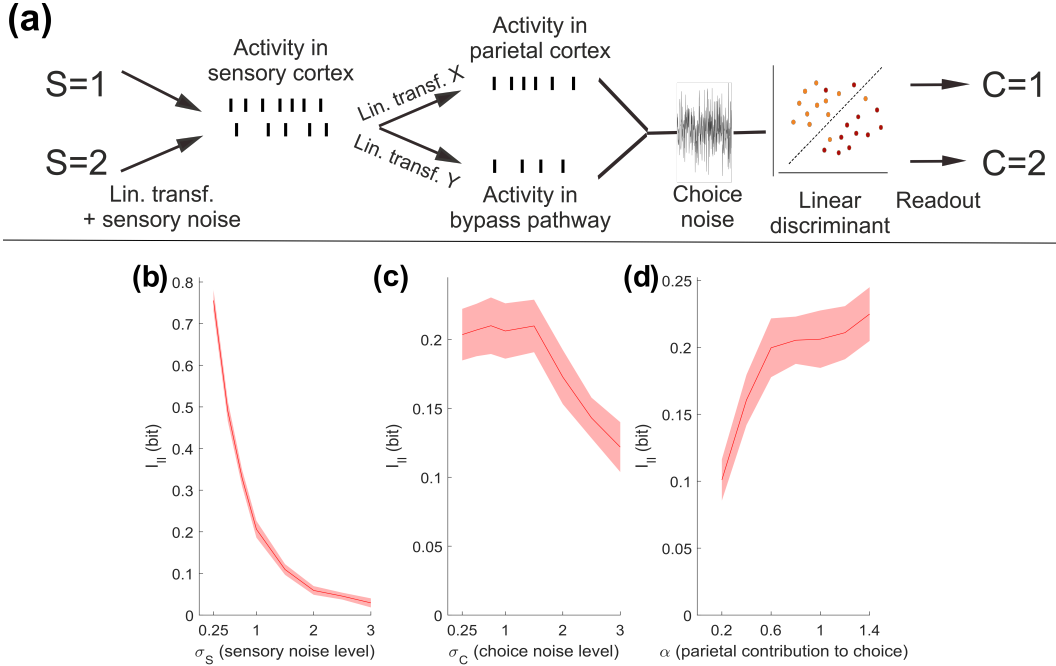

Figure 3: **a**) Schematics of the simulated model used to test our framework. In each trial, a binary stimulus is linearly converted into a "sensory-cortex activity" after the addition of 'sensory noise'. This signal is then separately converted to two higher-level activities, namely a "parietal-cortex activity" $R$ and a "bypass-pathway activity" $R'$. $R$ and $R'$ are then combined with parametrically tunable weights and, after the addition of "choice noise", this signal is fed to a linear discriminant. The output of the discriminant, that is the decoded stimulus $\hat{s}$, drives the binary choice $c$. We computed the intersection information of $R$ to extract the part of the stimulus information encoded in the "parietal cortex" that contributes to the final choice. **b-d**) Intersection Information for the simulations represented in **a**). Mean $\pm$ sem of $I_{\mathrm{II}}(S;R;C)$ across 100 experimental sessions, each relying on 100 simulated trials, as a function of three independently varied simulation parameters. **b**) Intersection Information decreases when the stimulus representation in the parietal cortex $R$ is more noisy (higher sensory noise $\sigma_S$ ). **c**) Intersection Information decreases when the beneficial contribution of the stimulus information carried by parietal cortex $R$ to the final choice is reduced by increasing choice noise $\sigma_C$. **d**) Intersection Information increases when the parietal cortex $R$ contributes more strongly to the final choice by increasing the parietal to bypass ratio $\alpha$.

In each simulated session, we estimated the joint probability $p_{\mathrm{session}}(s, r, c)$ of the stimulus $S$, the response in parietal cortex $R$, and the choice $C$, from 100 simulated trials. We computed, separately for each simulated session, an intersection information $I_{\mathrm{II}}(S;R;C)$ value from the estimated $p_{\mathrm{session}}(s, r, c)$. Here, and in all the analyses presented throughout the paper, we used a quadratic extrapolation procedure to correct for the limited sampling bias of information [28]. In Fig.3b-d we show mean $\pm$ s.e.m. of $I_{\mathrm{II}}(S;R;C)$ values across 100 independent experimental sessions, as a function of each of the three simulation parameters.

We found that $I_{\mathrm{II}}(S;R;C)$ decreases with increasing $\sigma_S$ (Fig.3b). This result was explained by the fact that increasing $\sigma_S$ reduces the amount of stimulus information that is passed to the simulated parietal activity $R$, and thus also reduces the portion of such information that can inform choice and can be used to perform the task appropriately. We found that $I_{\mathrm{II}}(S;R;C)$ decreases with increasing $\sigma_C$ (Fig.3c), consistently with the intuition that for higher values of $\sigma_C$ the choice depends more weakly on the activity of the simulated parietal activity $R$, which in turn also reduces how accurately the choice reflects the stimulus in each trial. We also found that $I_{\mathrm{II}}(S;R;C)$ increases with increasing $\alpha$ (Fig.3d), because when $\alpha$ is larger the portion of stimulus information carried by the simulated parietal activity $R$ that benefits the behavioral performance is larger.

# 4 Using our measure to rank candidate neural codes for task performance: studying the role of spike timing for somatosensory texture discrimination

The neural code was traditionally defined in previous studies as the set of features of neural activity that carry all or most sensory information. In this section, we show how $I_{\mathrm{II}}(S; R; C)$ can be used to quantitatively redefine the neural code as the set of features that contributes the most sensory information for task performance. The experimenter can thus use $I_{\mathrm{II}}(S; R; C)$ to rank a set of candidate neural features $\{R_1, ..., R_N\}$ according to the numerical ordering $I_{\mathrm{II}}(S; R_{i_1}; C) \leq ... \leq I_{\mathrm{II}}(S; R_{i_N}; C)$. An advantage of the information-theoretic nature of $I_{\mathrm{II}}(S; R; C)$ is that it quantifies intersection information on the meaningful scale of bits, and thus enables a quantitative comparison of different candidate neural codes. If for example $I_{\mathrm{II}}(S; R_1; C) = 2I_{\mathrm{II}}(S; R_2; C)$ we can quantitatively interpret that the code $R_1$ provides twice as much information for task performance as $R_2$. This interpretation is not as meaningful, for example, when comparing different values of fraction-correct measures [17].

To illustrate the power of $I_{\mathrm{II}}(S; R; C)$ for evaluating and ranking candidate neural codes, we apply it to real data to investigate a fundamental question: is the sensory information encoded in millisecond-scale spike times used by the brain to perform perceptual discrimination? Although many studies have shown that millisecond-scale spike times of cortical neurons encode sensory information not carried by rates, whether or not this information is used has remained controversial [16, 29, 30]. It could be, for example, that spike times cannot be read out because the biophysics of the readout neuronal systems is not sufficiently sensitive to transmit this information, or because the readout neural systems do not have access to a stimulus time reference that could be used to measure these spike times [31].

To investigate this question, we used intersection information to compute whether millisecond-scale spike timing of neurons (n=299 cells) in rat primary (S1) somatosensory cortex provides information that is used for performing a whisker-based texture discrimination task (Figure 4a-b). Full experimental details are reported in [32]. In particular, we compared $I_{\mathrm{II}}(S; \text{timing}; C)$ with the intersection information carried by rate $I_{\mathrm{II}}(S; \text{rate}; C)$, i.e. information carried by spike counts over time scales of tens of milliseconds. We first computed a spike-timing feature by projecting the single-trial spike train onto a zero-mean timing template (constructed by linearly combining the first three spike trains PCs to maximize sensory information, following the procedure of [32]), whose shape indicated the weight assigned to each spike depending on its timing (Figure 4a). Then we computed a spike-rate feature by weighting the spikes with a flat template which assigns the same weight to spikes independently of their time. Note that this definition of timing, and in particular the fact that the timing template was zero mean, ensured that the timing variable did not contain any rate information. We verified that this calculation provided timing and rate features that had negligible ($-0.0030 \pm 0.0001$ across the population) Pearson correlation.

The difficulty of the texture discrimination task was set so that the rat learned the task well but still made a number of errors in each session (mean behavioral performance 76.9%, p<0.001 above chance, paired t-test). These error trials were used to decouple in part choice from stimulus coding and to assess the impact of the sensory neural codes on behavior by computing intersection information. We thus computed information across all trials, including both behaviorally correct and incorrect trials. We found that, across all trials and on average over the dataset, timing carried similar texture information to rate (Figure 4b) ($(9 \pm 2) \times 10^{-3}$ bit in timing, $(8.5 \pm 1.1) \times 10^{-3}$ bit in rate, p=0.78 two-sample t-test), while timing carried more choice information than rate ($(16 \pm 1) \times 10^{-3}$ bit in timing, $(3.0 \pm 0.7) \times 10^{-3}$ bit in rate, p<$10^{-15}$ two-sample t-test). If we used only traditional measures of stimulus and choice information, it would be difficult to decide which code is most helpful for task performance. However, when we applied our new information-theoretic framework, we found that the intersection information $I_{\mathrm{II}}$ (Figure 4b) was higher for timing than for rate ($(7 \pm 1) \times 10^{-3}$ bit in timing, $(3.0 \pm 0.6) \times 10^{-3}$ bit in rate, p<0.002 two-sample t-test), thus suggesting that spike timing is a more crucial neural code for texture perception than spike rate.

Interestingly, intersection information $I_{\mathrm{II}}$ was approximately 80% of the total sensory information for timing, while it was only 30% of the total sensory information for rate. This suggests that in somatosensory neurons timing information about the texture is read out, and influences choice, more efficiently than rate information, contrarily to what is widely assumed in the literature [34]. These results confirm early results that were obtained with a decoding-based intersection information measure [32]. However, the information theoretic results in Fig.4b have the advantage that they do

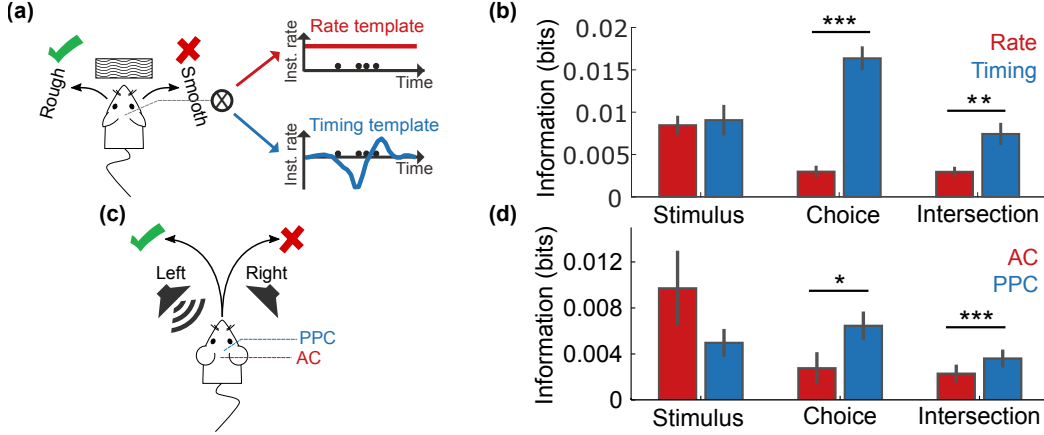

Figure 4: Intersection Information for two experimental datasets. **a**: Simplified schematics of the experimental setup in [32]. Rats are trained to distinguish between textures with different degrees of coarseness (left), and neural spiking data from somatosensory cortex (S1) is decomposed in independent *rate* and *timing* components (right). **b**: Stimulus, choice and intersection information for the data in panel **a**. Spike timing carries as much sensory information (p=0.78, 2-sample t-test), but more choice information (p<$10^{-15}$), and more $I_{II}$ (p<0.002) than firing rate. **c**: Simplified schematics of the experimental setup in [33]. Mice are trained to distinguish between auditory stimuli located to their left or to their right. Neural activity is recorded in auditory cortex (AC) and posterior parietal cortex (PPC) with 2-photon calcium imaging. **d**: Stimulus, choice and intersection information for the data in panel **c**. Stimulus information does not differ significantly between AC and PPC, but PPC has more choice information (p<0.05) and more $I_{II}$ than AC (p<$10^{-6}$, 2-sample t-test).

not depend on the use of a specific decoder to calculate intersection information. Importantly, the new information theoretic approach also allowed us to quantify the proportion of sensory information in a neural code that is read out downstream for behavior, and thus to obtain the novel conclusion that only spike timing is read out with high efficiency.

## 5 Application of intersection information to discover brain areas transforming sensory information into choice

Our intersection information measure $I_{II}(S; R; C)$ can also be used as a metric to discover and index brain areas that perform the key computations needed for perceptual discrimination, and thus turn sensory information into choice. Suppose for example that we are investigating this issue by recording from populations of neurons in different areas. If we rank the neural activities in the recorded areas according to the sensory information they carry, we will find that primary sensory areas are ranked highly. Instead, if we rank the areas according to the choice information they carry, the areas encoding the motor output will be ranked highly. However, associative areas that transform sensory information into choice will not be found by any of these two traditional sensory-only and choice-only rankings, and there is no currently established metric to quantitatively identify such areas. Here we argue that $I_{II}(S; R; C)$ can be used as such metric.

To illustrate this possible use of $I_{II}(S; R; C)$, we analyzed the activity of populations of single neurons recorded in mice with two-photon calcium imaging either in Auditory Cortex (AC, n=329 neurons) or in Posterior Parietal Cortex (PPC, n=384 neurons) while the mice were performing a sound location discrimination task and had to report the perceived sound location (left vs right) by the direction of their turn in a virtual-reality navigation setup (Fig.4c; full experimental details are available in Ref.[33]). AC is a primary sensory area, whereas PPC is an association area that has been described as a multisensory-motor interface [35, 36, 37], was shown to be essential for virtual-navigation tasks [36], and is implicated in the spatial processing of auditory stimuli [38, 39].

When applying our information theoretic formalism to these data, we found that similar stimulus (sound location) information was carried by the firing rate of neurons in AC and PPC (AC: $(10 \pm 3) \times 10^{-3}$ bit, PPC: $(5 \pm 1) \times 10^{-3}$ bit, p=0.17, two-sample t-test). Cells in PPC carried

more choice information than cells in AC (AC: $(2.8 \pm 1.4) \times 10^{-3}$ bit, PPC: $(6.4 \pm 1.2) \times 10^{-3}$ bit, p<0.05, two-sample t-test). However, neurons in PPC had values of $I_{\mathrm{II}}$ ($(3.6 \pm 0.8) \times 10^{-3}$ bit) higher (p<$10^{-6}$, two-sample t-test) than those of AC ($(2.3 \pm 0.8) \times 10^{-3}$ bit): this suggests that the sensory information in PPC, though similar to that of AC, is turned into behavior into a much larger proportion (Figure 4d). Indeed, the ratio between $I_{\mathrm{II}}(S; R; C)$ and sensory information was higher in PPC than in AC (AC: $(24 \pm 11)\,\%$, PPC: $(73 \pm 24)\,\%$, p<0.03, one-tailed z-test). This finding reflects the associative nature of PPC as a sensory-motor interface. This result highlights the potential usefulness of $I_{\mathrm{II}}(S; R; C)$ as an important metric for the analysis of neuro-imaging experiments and the quantitative individuation of areas transforming sensory information into choice.

## 6  Discussion

Here, we derived a novel information theoretic measure $I_{\mathrm{II}}(S; R; C)$ of the behavioral impact of the sensory information carried by the neural activity features $R$ during perceptual discrimination tasks. The problem of understanding whether the sensory information in the recorded neural features really contributes to behavior is hotly debated in neuroscience [16, 17, 30]. As a consequence, a lot of efforts are being devoted to formulate advanced analytical tools to investigate this question [17, 40, 41]. A traditional and fruitful approach has been to compute the correlation between trial-averaged behavioral performance and trial-averaged stimulus decoding when presenting stimuli of increasing complexity [13, 14, 18]. However, this measure does not capture the relationship between fluctuations of neural sensory information and behavioral choice in the same experimental trial. To capture this single-trial relationship, Ref.[17] proposed to use a specific stimulus decoding algorithm to classify trials that give accurate sensory information, and then quantify the increase in behavioral performance in the trials where the sensory decoding is correct. However, this approach makes strong assumptions about the decoding mechanism, which may or may not be neurally plausible, and does not make use of the full structure of the trivariate $S, R, C$ dependencies.

In this work, we solved all the problems described above by extending the recent Partial Information Decomposition framework [19, 20] for the analysis of trivariate dependencies to identify $I_{\mathrm{II}}(S; R; C)$ as a part of the redundant information about $C$ shared between $S$ and $R$ that is also a part of the sensory information $I(S : R)$. This quantity satisfies several essential properties of a measure of intersection information between the sensory coding $s \to r$ and the consequent behavioral readout $r \to c$, that we derived from the conceptual notions elaborated in Ref.[17]. Our measure $I_{\mathrm{II}}(S; R; C)$ provides a single-trial quantification of how much sensory information is used for behavior. This quantification refers to the absolute physical scale of bit units, and thus enables a direct comparison of different candidate neural codes for the analyzed task. Furthermore, our measure has the advantages of information-theoretical approaches, that capture all statistical dependencies between the recorded quantities irrespective of their relevance to neural function, as well as of model-based approaches, that link directly empirical data with specific theoretical hypotheses about sensory coding and behavioral readout but depend strongly on their underlying assumptions (see e.g. [12]).

An important direction for future expansions of this work will be to combine $I_{\mathrm{II}}(S; R; C)$ with interventional tools on neural activity, such as optogenetics. Indeed, the novel statistical tools in this work cannot distinguish whether the measured value of intersection information $I_{\mathrm{II}}(S; R; C)$ derives from the causal involvement of $R$ in transmitting sensory information for behavior, or whether $R$ only correlates with causal information-transmitting areas [17].

More generally, this work can help us mapping information flow and not only information representation. We have shown above how computing $I_{\mathrm{II}}(S; R; C)$ separates the sensory information that is transmitted downstream to affect the behavioral output from the rest of the sensory information that is not transmitted. Further, another interesting application of $I_{\mathrm{II}}$ arises if we replace the final choice $C$ with other nodes of the brain networks, and compute with $I_{\mathrm{II}}(S; R_1; R_2)$ the part of the sensory information in $R_1$ that is transmitted to $R_2$. Even more generally, besides the analysis of neural information processing, our measure $I_{\mathrm{II}}$ can be used in the framework of network information theory: suppose that an input $X = (X_1, X_2)$ (with $X_1 \perp\!\!\!\perp X_2$) is encoded by 2 different parallel channels $R_1, R_2$, which are then decoded to produce collectively an output $Y$. Suppose further that experimental measurements in single trials can only determine the value of $X$, $Y$, and $R_1$, while the values of $X_1, X_2, Y_1, Y_2, R_2$ are experimentally inaccessible. As we show in Supp. Fig. 3, $I_{\mathrm{II}}(X; R_1; Y)$ allows us to quantify the information between $X$ and $Y$ that passes through the channel $R_1$, and thus does not pass through the channel $R_2$.

# 7 Acknowledgements and author contributions

GP was supported by a Seal of Excellence Fellowship CONISC. SP was supported by Fondation Bertarelli. CDH was supported by grants from the NIH (MH107620 and NS089521). CDH is a New York Stem Cell Foundation Robertson Neuroscience Investigator. TF was supported by the grants ERC (NEURO-PATTERNS) and NIH (1U01NS090576-01). CK was supported by the European Research Council (ERC-2014-CoG; grant No 646657).

Author contributions: SP, GP and EP conceived the project; GP and EP performed the project; CAR, MED and CDH provided experimental data; GP, EP, HS, CK, SP and TF provided materials and analysis methods; GP, EP and SP wrote the paper; all authors commented on the manuscript; SP supervised the project.

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
