[Supplementary Material · supplemental-material-camera-ready.pdf]

# Supplementary information for *Quantifying how much sensory information in a neural code is relevant for behavior*

**Giuseppe Pica**[1,2]
giuseppe.pica@iit.it

**Eugenio Piasini**[1]
eugenio.piasini@iit.it

**Houman Safaai**[1,3]
houman_safaai@hms.harvard.edu

**Caroline A. Runyan**[3,4]
runyan@pitt.edu

**Mathew E. Diamond**[5]
diamond@sissa.it

**Tommaso Fellin**[2,6]
tommaso.fellin@iit.it

**Christoph Kayser**[7,8]
christoph.kayser@uni-bielefeld.de

**Christopher D. Harvey**[3]
Christopher_Harvey@hms.harvard.edu

**Stefano Panzeri**[1,2]
stefano.panzeri@iit.it

[1] Neural Computation Laboratory, Center for Neuroscience and Cognitive Systems@UniTn,
Istituto Italiano di Tecnologia, Rovereto (TN) 38068, Italy
[2] Neural Coding Laboratory, Center for Neuroscience and Cognitive Systems@UniTn,
Istituto Italiano di Tecnologia, Rovereto (TN) 38068, Italy
[3] Department of Neurobiology, Harvard Medical School, Boston, MA 02115, USA
[4] Department of Neuroscience, University of Pittsburgh,
Center for the Neural Basis of Cognition, Pittsburgh, USA
[5] Tactile Perception and Learning Laboratory,
International School for Advanced Studies (SISSA), Trieste, Italy
[6] Optical Approaches to Brain Function Laboratory,
Istituto Italiano di Tecnologia, Genova 16163, Italy
[7] Institute of Neuroscience and Psychology, University of Glasgow, Glasgow, UK
[8] Department of Cognitive Neuroscience, Faculty of Biology,
Bielefeld University, Universitätsstr. 25, 33615 Bielefeld, Germany

## 1   Properties of $I_{\text{II}}$

In this section we prove the properties of $I_{\text{II}}(S; R; C)$ as discussed in Section 2 in the main text. We first prove the properties 1-3 at the same time:

$$I_{\text{II}}(S; R; C) \leq I(S : R), I(R : C), I(S : C). \tag{1}$$

*Proof.* From the definition in Eq.3 in the main text, $I_{\text{II}}(S; R; C) = \min\{SI(C : \{S; R\}), SI(S : \{R; C\})\}$, but $SI(C : \{S; R\}) \leq I(S : C), I(R : C)$ [1, 2] and similarly $SI(S : \{R; C\}) \leq I(S : R), I(S : C)$. Thus, $I_{\text{II}}(S; R; C) \leq SI(C : \{S; R\}), SI(S : \{R; C\}) \leq I(S : R), I(R : C), I(S : C)$. $\qquad\square$

Note that, as a corollary of $I_{\text{II}}(S; R; C) \leq I(S : R)$ (Eq.1), we find that $S \perp\!\!\!\perp R \implies I_{\text{II}}(S; R; C) = 0$, as reported in Section 2 in the main text.

Then we prove property 4:

$$I_\mathrm{II}(S; \{R_1, R_2\}; C) \geq I_\mathrm{II}(S; R_1; C), I_\mathrm{II}(S; R_2; C). \tag{2}$$

*Proof.* Both $SI(C : \{S; (R_1, R_2)\}) \geq SI(C : \{S; R_1\}), SI(C : \{S; R_2\})$ [3] and, similarly, $SI(S : \{C; (R_1, R_2)\}) \geq SI(S : \{C; R_1\}), SI(S : \{C; R_2\})$. Thus, $I_\mathrm{II}(S; R_1; C) = \min[SI(C : \{S; R_1\}), SI(S : \{R_1; C\})] \leq \min[SI(C : \{S; (R_1, R_2)\}), SI(S : \{C; (R_1, R_2)\})] = I_\mathrm{II}(S; \{R_1, R_2\}; C)$. The same argument can be repeated exactly if we swap $R_1 \leftrightarrow R_2$. □

## 2  Further examples where $I_{II} = 0$

In this section, we develop in detail two examples where the intersection information $I_\mathrm{II}(S; R; C)$ is zero, but this property can't be immediately determined from the independence relationships between the stimulus $S$, the neural response $R$, and the choice $C$. In other words, the intersection information is zero even though this can't be read off a representation of $p(s, r, c)$ as a probabilistic graphical model.

### 2.1  The choice $C$ can be independent of the stimulus $S$ even in presence of a neural code $R$ which is informative about both $S$ and $C$

Consider a behavioural task where the animal needs to recognise one out of a set of 4 (equiprobable) distinct sensory stimuli. As a concrete example, take a visual stimulus where one quadrant of a screen is filled black:

$$s_{00} = \blacksquare \quad , \quad s_{01} = \blacksquare$$
$$s_{10} = \blacksquare \quad , \quad s_{11} = \blacksquare$$

and the animal can make 1 out of 4 distinct choices to report the stimulus identity:

$$c_{00} = \nwarrow \quad , \quad c_{01} = \nearrow$$
$$c_{10} = \swarrow \quad , \quad c_{11} = \searrow$$

The behavioural performance would be perfect if each choice were exactly associated with the corresponding stimulus, such as $\blacksquare \leftrightarrow \nwarrow$, $\blacksquare \leftrightarrow \nearrow$, and so on. Finally, suppose that during the experiment we record the activity of one neuron, and that this activity assumes one of 4 possible states:

$$r_{00} = \text{|||\;|}$$
$$r_{01} = \text{||\;\;\;|}$$
$$r_{10} = \text{|\;\;||\;|}$$
$$r_{11} = \text{|\;\;\;|\;\;\;|}$$

This is a temporal code where each state can be described with the duration of two inter-spike intervals.

We will now model a case in which the neural code is informative about both the stimulus and the behaviour, but the stimulus is not informative about the behaviour. In other words, a case in which *the behaviour depends on the code and the code depends on the stimulus, but the behaviour doesn't depend on the stimulus.* Assume that the first inter-spike interval of the neural code depends on the vertical component of the stimulus, while the duration of the second inter-spike interval is set at random (Figure 1b). Suppose also that one component of the behaviour – say the horizontal one – is determined by the duration of the second inter-spike interval, but the other is again set at random (Figure 1c). This can be represented by a diagram like that in Figure 1a.

The end result is that in this model we have

$$H(S) = H(R) = H(C) = 2\,\text{bits}$$
$$I(S : R) = I(R : C) = 1\,\text{bit}$$
$$I(S : C) = 0$$
$$I_\mathrm{II}(S; R; C) = 0.$$

Figure 1: a: graphical representation of the joint probability distribution $p(s, r, c)$ for an example where the neural response is informative about the stimulus and the choice, but the choice may be unrelated to the stimulus. The probabilistic graphical model representation is augmented by a color code illustrating the nature of the information carried by statistical relationships between variables. Red: information about the stimulus; blue: information about anything else (internal noise, distractors, etc). b: Conditional probability table of the recoded spiking activity given the sensory stimulus. The duration of the first inter-spike interval is determined by the vertical component of the stimulus, while the duration of the second inter-spike interval is independent of the stimulus and assumes one of the two possible values ("long" and "short") with equal probability. c: Conditional probability table for the choice given the neural activity. The horizontal component of the behaviour variable depends on the duration of the second inter-spike interval in the neural activity, and the vertical component of the behaviour variable assumes one of the two possible values with equal probability.

In conclusion, this is an example of a case where a neural code can have significant stimulus and choice information, but it does not contribute at all to the performing of the sensory discrimination task: indeed, the stimulus and the behaviour are statistically independent (the animal performs the task at chance level). In other words, the features of the neural code that tell us something about the stimulus are completely independent from the features of the code that affect the behaviour. As expected, it is found that $I_{\Pi}(S; R; C) = 0$ in this scenario.

### 2.1.1 Stimulus, neural code and choice all being informative about each other does not mean that the information carried by the neural code about $S$ and $C$ is relevant for the behavioural task

Consider the same type of experimental setup as in the previous example. We will now model a case in which:

- the animal can differentiate perfectly the stimuli on the horizontal axis but not on the vertical axis (i.e. the horizontal component of the choice depends on the horizontal component of the stimulus);

- the neural code depends on the vertical component of the stimulus;

- the vertical component of the choice depends on the neural code, but in a way which is unrelated to the stimulus.

To do this, we assume first of all that $R$ depends on $S$ through the conditional probabilities given in Figure 2c; this means that the duration of the first inter-spike interval of $R$ is controlled by whether the stimulus is in the upper or lower half of the screen, but the duration of the second inter-spike interval is not informative about the stimulus. We also assume that $C$ is completely determined by $S$

(a)

| $s =$ | ◩ | ⊟ | ◧ | ◨ |
|---|---|---|---|---|
| $p(c = \nwarrow \mid s)$ | 0.5 | 0.5 | 0 | 0 |
| $p(c = \nearrow \mid s)$ | 0.5 | 0.5 | 0 | 0 |
| $p(c = \searrow \mid s)$ | 0 | 0 | 0.5 | 0.5 |
| $p(c = \nearrow \mid s)$ | 0 | 0 | 0.5 | 0.5 |

(b)

| $s =$ | ◱ | ◩ | ◧ | ◨ |
|---|---|---|---|---|
| $p(r = \text{⊔⊔⊔\_\_} \mid s)$ | 0.5 | 0 | 0 | 0.5 |
| $p(r = \text{⊔⊔\_⊔\_} \mid s)$ | 0.5 | 0 | 0 | 0.5 |
| $p(r = \text{⊔\_⊔⊔\_} \mid s)$ | 0 | 0.5 | 0.5 | 0 |
| $p(r = \text{⊔\_\_⊔⊔} \mid s)$ | 0 | 0.5 | 0.5 | 0 |

(c)

| $r =$ | ⊔⊔⊔\_ | ⊔⊔\_⊔ | ⊔\_⊔⊔ | ⊔\_\_⊔ |
|---|---|---|---|---|
| $p(c = \nwarrow \mid r)$ | 0.5 | 0 | 0 | 0.5 |
| $p(c = \nearrow \mid r)$ | 0.5 | 0 | 0 | 0.5 |
| $p(c = \searrow \mid r)$ | 0 | 0.5 | 0.5 | 0 |
| $p(c = \nearrow \mid r)$ | 0 | 0.5 | 0.5 | 0 |

(d)

Figure 2: a: graphical representation of the joint probability distribution $p(s, r, c)$ for an example where stimulus, neural activity and choice are all informative about each other, but the neural activity has no intersection information. The probabilistic graphical model representation is augmented by a color code illustrating the nature of the information carried by statistical relationships between variables. Red: information about the stimulus; blue: information about anything else (internal noise, distractors, etc). b: Conditional probability table for the behaviour given the stimulus. This corresponds to the animal being able to differentiate stimuli on the horizontal axis, but not on the vertical axis. c: Conditional probability table of the recoded spiking activity given the sensory stimulus. Note that the neural activity only allows to distinguish between different stimuli on the vertical axis. d: Conditional probability table for the behaviour given the neural activity. This corresponds to the behaviour depending on whether the second inter-spike interval has the same duration of the first one, but not on the duration of the first spike interval itself.

and $R$ taken together in the following way:

$$c(s_{00}, r_{00}) = c(s_{00}, r_{11}) = c(s_{10}, r_{00}) = c(s_{10}, r_{11}) = c_{00}$$
$$c(s_{00}, r_{01}) = c(s_{00}, r_{10}) = c(s_{10}, r_{01}) = c(s_{10}, r_{10}) = c_{01}$$
$$c(s_{01}, r_{01}) = c(s_{01}, r_{10}) = c(s_{11}, r_{01}) = c(s_{11}, r_{10}) = c_{10}$$
$$c(s_{01}, r_{00}) = c(s_{01}, r_{11}) = c(s_{11}, r_{00}) = c(s_{11}, r_{11}) = c_{11}$$

We can then derive the conditional probabilities of $C$ given $S$ and $C$ given $R$, reported respectively in Figure 2b and Figure 2d. From these, we can see that, as required, the horizontal component of $C$ depends directly on the horizontal component of $S$, and the vertical component of $C$ depends on whether the duration of the second inter-spike interval in $R$ matches that of the first interval – but this dimension of the neural code is independent of the sensory stimulus $R$. As expected, it is found that $I_{\mathrm{II}}(S; R; C) = 0$ in this scenario.

The end result is that, in this model, we have:

$$H(S) = H(R) = H(C) = 2 \, \mathrm{bit}$$
$$I(S : R) = I(S : C) = I(R : C) = 1 \, \mathrm{bit}$$
$$I_{\mathrm{II}}(S; R; C) = 0,$$

i.e. the information that the neural code carries about the stimulus *is not relevant* for the behaviour, and conversely the features of the code that affect the behaviour are unrelated to the sensory stimulus.

# 3  Details of the numerical calculation of $I_{\Pi}$

Our novel measure of intersection information $I_{\Pi}(S; R; C)$ relies on the definitions of the Partial Information Decomposition (PID) atoms as proposed in [2] (see Eq.3 in the main text). Suppose that the stimulus $s$ can take values in a discrete set $s \in \mathcal{S}$, the response in a discrete set $r \in \mathcal{R}$, and the choice in a discrete set $c \in \mathcal{C}$. Once the trivariate probability distribution $p(s, r, c)$ has been estimated from experimental data, the PID that decomposes the information that $S$ and $R$ carry about $C$ requires the definition of the space $\Delta_p$ of all trivariate distributions $q(s, r, c)$ in the following set:

$$\Delta_p = \{q(s, r, c) : q(s, c) = p(s, c) \text{ and } q(r, c) = p(r, c), \ \forall s \in \mathcal{S}, r \in \mathcal{R}, c \in \mathcal{C}\}. \tag{3}$$

In words, to decompose the information that two variables (sources) carry about a third variable (target), Bertschinger *et al.* proposed to consider the space $\Delta_p$ of all probability distributions $q(\text{Source 1}, \text{Source 2}, \text{Target})$ with the same pairwise marginal distributions on the pairs (Source 1, Target), (Source 2, Target) as the original $p(\text{Source 1}, \text{Source 2}, \text{Target})$.

In particular, the redundant information that $S$ and $R$ share about $C$, $SI(C : \{S; R\})$, is defined as the solution of the following convex optimization problem on the space $\Delta_p$ [2]:

$$SI(C : \{S; R\}) \equiv \max_{q \in \Delta_p} CoI_q(S; R; C), \tag{4}$$

where $CoI_q(S; R; C) \equiv I_q(S : R) - I_q(S : R|C)$ is the co-information corresponding to the probability distribution $q(s, r, c)$. Ref.[2] discussed how the space $\Delta_p$ is a polytope whose elements can be parametrized in terms of real linear combinations of $|\mathcal{C}|(|\mathcal{S}| - 1)(|\mathcal{R}| - 1)$ vectors [4], where $|\mathcal{X}|$ is the number of elements in the set $\mathcal{X}$. The problem in Eq.4 can thus be treated as a convex optimization problem on a linear vector space with finite dimensions. The objective function of this problem, $CoI_q(S; R; C)$, is a highly non-linear (convex) function of the elements of the parametrization space, which is compact and convex. To solve this problem, we adopted the Frank-Wolfe optimization algorithm, also known as the conditional gradient method [5].

Open-source software to calculate our measure of intersection information $I_{\Pi}(S; R; C)$ from any finite-dimensional probability distribution $p(s, r, c)$ estimated from experimental or simulated data is available at this link: https://doi.org/10.5281/zenodo.850362.

# 4  Details about the simulations of the sensory discrimination task

The simulations described in Section 3 of the main text (see also Fig.2a therein) were implemented as follows. In every simulated trial, the simulated neural activity in sensory cortex was a continuous variable $SC$ defined as

$$SC = s + SN, \tag{5}$$

where $s$ was a randomly drawn binary stimulus ($s \in \{s_1, s_2\}$) and the sensory noise SN $\sim \mathcal{N}(0, \sigma_S)$ was normally distributed with tunable variance $\sigma_S$. Four independent values of $SC$, corresponding to four independent values of SN but to the same value of $s$, $\{SC(i)\}_{i=1:4}$, were drawn for each trial. The simulated activity in parietal cortex ($R$) and the simulated activity in the bypass pathway ($R'$) were determined, in each trial, as follows:

$$R = \begin{cases} SC(1) + SC(2) & \text{if} \quad s = s_1 \\ SC(1) - SC(2) & \text{if} \quad s = s_2 \end{cases} \tag{6}$$

$$R' = \begin{cases} SC(3) + SC(4) & \text{if} \quad s = s_1 \\ SC(3) - SC(4) & \text{if} \quad s = s_2 \end{cases} \tag{7}$$

We then combined $R$ and $R'$ into a unique continuous variable $R_{\text{higher}}$, as follows:

$$R_{\text{higher}} = R + \frac{1}{\alpha} R', \tag{8}$$

i.e. the ratio between the $R$-weight and the $R'$-weight is indicated with the coefficient $\alpha$. The variable $R_{\text{higher}} + CN$, where the choice noise $CN \sim \mathcal{N}(0, \sigma_C)$ was normally distributed with standard deviation $\sigma_C$, was finally fed to a linear discriminant, whose outcome $\hat{s}$ determined the binary choice $c$. The linear discriminant was based on a multivariate regularized linear discriminant analysis [6, 7], that was applied to the input of the decoder to derive an optimal linear separation between two groups of trials: one corresponding to a decoded stimulus $\hat{s} = s_1$, the other to $\hat{s} = s_2$ (the regularization parameter was fixed to 0.1).

$$I_{\mathrm{II}}(R_1) = I(X_1 : Y_1)$$

Figure 3: In the main text, $I_{\mathrm{II}}(S; R; C)$ is defined and characterized as a measure of intersection information in neuroscience. However, the definition in Eq.3 in the main text does not rely on any specific assumption about the nature of the statistical dependencies that underlie a trivariate system. In other words, $I_{\mathrm{II}}$ can still be defined for three generic variables $(X, R, Y)$. To give further intuition about what $I_{\mathrm{II}}$ quantifies, we considered a system where a stochastic input variable $X = (X_1, X_2)$ is fed to two distinct encoding channels with outputs $R_1, R_2$; importantly, the inputs to the channels are independent, $X_1 \perp\!\!\!\perp X_2$. The variables $R_1, R_2$ are then independently decoded to produce collectively a joint output $Y = (Y_1, Y_2)$. We further assumed that only the black variables in the scheme could be measured, while the red variables could not. Thus, we assumed that only the joint distribution $p(x, r_1, y)$ was known. Following the definition in Eq.3 in the main text, $I_{\mathrm{II}}(X; R_1; Y)$ equals the Reversible Shared Information $RSI(X \overset{R_1}{\leftrightarrow} Y)$ that we defined in Ref.[8]. As we showed in that work, $I_{\mathrm{II}}(X; R_1; Y)$ then quantifies the information between $X$ and $Y$ that passes through $R_1$, i.e. $I_{\mathrm{II}}(X; R_1; Y) = I(X_1 : Y_1)$. Note that, without computing the intersection information $I_{\mathrm{II}}(X; R_1; Y)$, we could not estimate $I(X_1 : Y_1)$ from the experimental measurements, since $p(x_1, y_1)$ is not known directly.

## 5 Experimental methods

### 5.1 Texture discrimination task

Experimental data was collected as described in [9]. Briefly, in that work, five Wistar rats were handled and habituated to the task for one week. Twenty-four hours prior to the onset of training, and throughout the training, rats were water-restricted but had free access to food in the home cage. On each trial, one out of four plates textured with varying degrees of smoothness was presented before the rat. The rat perched on the front edge of a rectangular platform and extended itself forward to contact the discriminandum with its whiskers. After palpating the texture, the rat withdrew and turned to either left or right to lick a drinking spout (Figure 4a, main text). Only if it approached the correct drinking spout was it given a water reward; for an incorrect choice, it received no water. Associations between texture and reward location were fixed for each animal but varied in the five rats. Neuronal responses were recorded from a 12-tetrode array. Spikes were sorted offline on the basis of the amplitude and waveform energy on each of the four channels of the tetrode. All experiments were conducted in accordance with the National Institute of Health and international standards for the care and use of animals. Protocols were approved by the SISSA Ethics Committee and were supervised by a consulting veterinarian.

### 5.2 Auditory discrimination task

Experimental data was collected as described in [10]. Briefly, a sound localization task was developed in which mice reported perceptual decisions by navigating through a visual virtual reality T-maze [11]. As mice ran down the T-stem, a sound cue was played from one of eight possible locations in head-centered, real-world coordinates. Mice reported whether the sound originated from their left or right by turning in that direction at the T-intersection. Populations of ~40–70 cells were simultaneously imaged (GCaMP6f) during each individual recording session. Temporal resolution of image acquisition was $60\,\mathrm{ms}$. Calcium traces were deconvolved to estimate the relative firing rate within each imaging frame [12]. For each cell, the total firing rate in a given trial was obtained by averaging the deconvolved calcium signal over the trial. All experimental procedures were approved

by the Harvard Medical School Institutional Animal Care and Use Committee and were performed in compliance with the Guide for Animal Care and Use of Laboratory Animals. Imaging data were collected from four male C57BL/6J mice (Jackson Labs) that were seven weeks old at the initiation of behavior task training.