[Reviews · NeurIPS 2017]

Reviewer 1



This manuscript proposes a new information-theoretic measure for quantifying the amount of information about the stimuli carried in the neural response, which is also used for behavior for a perceptual discrimination tasks. This work builds on the previous literature of Partial Information Decomposition. The information-theoretic measure is tested using a simulation and two published experimental datasets. Overall, I find that this is a potentially interesting contribution. The manuscript is generally well-written, except it misses the definitions of a few key quantities. The general approach the authors use is principled and potentially lead to better way to examine the relation between the neural response, behavior and the stimuli. I have a few general as well as technical concerns about the applicability of this measure in typical experiments which I will detail below. Major concerns: Eq.(1) is a key equation, however, the terms at the r.h.s are not defined, and only intuitions are given. For example, how to compute SI(C:{S,R})? I think it would be useful to have that spell out explicitly. Minor comment- there is a typo in the last two terms, and now they appear to be identical. Eq. (2) gives the definition of the proposed new measure. However, what is the difference between this measure, and the bivariate redundancy measure proposed in Harder et al. (PRE, 2013; reference (21) in the submission)? In Harder et al., a measure with similar structure is proposed, see their Eq. (13). It is not clear to me how the "new definition" the authors proposed here is different from the Eq. (13) there. Many of the properties of the measure proposed here seems to be shared with the measure defined there. However, since the definition of SI(C:{S,R}) is not given in the current paper, I might be missing something important here. In any case, I'd like to understand whether the measure defined in Eq. (2) is indeed different from Eq. (13) of Harder et al (2013), as this seems to be critical for judging the novelty of the current paper. Regardless of whether the measure the authors proposed is indeed novel, there is a crucial question in terms of practically how useful this method is. I'd think one of the main challenges for using this method is estimating the joint probably of the stimulus, activity and the choice. But in the Supplementary Information, the authors only explained how to proceed after assuming this joint probability has been estimated. Do the authors make any important assumptions to simplify the computations here, in particular for the neural response R? For example, is the independence across neurons assumed? Furthermore, in many tasks with complex stimuli such as in many vision experiments, I'd think it is very difficult to estimate even the prior distribution on the stimulus, not to say the joint probability of the stimulus, the activity and the behavior. I understand that the authors here only focus on simple choice tasks in which the stimuli are modeled as binary variables, but the limitations of this approach should be properly discussed. In the experimental section, it seems that the number information is a bit small, e.g., ~10^{-3} bit. Can these number map to the values of choice probability and are they compatible? In general, it might be useful to discuss the relation of this approach to the approach of using choice probability, both in terms of the pros and the cons, as well as in which cases they relate to each other. In summary, my opinion is that the theoretical aspects of the paper seems to be incremental. However, I still find the paper to be of potential interests to the neurophysiologists. I have some doubts on the general applicability of the methods, as well as the robustness of the conclusions that can be drawn from this method, but still this paper may represent an interesting direction and may prove useful for studying the neural code of choice behavior. Minor: In eq. (2), the notation for I_{II}(R) is confusing, because it doesn't contain (R,C), but nonetheless it depends on them. This further leads to the confusion for I_{II}(R1,R2) at the top of line 113. What does it mean?

Reviewer 2



The paper proposed a novel of information-theoretic criterion to describe the intersection information between sensory coding and behavioral readout. Compared to previous studies, this criterion doesn’t require specific choice of decoding algorithm and considers the full structure in the measured statistical relationships between stimulus, responses and choice. The paper starts from the Partial Information Decomposition(PID), claiming that none of these components fit the notion of intersection information. Then it defines the new criterion to be the minimum of two shared informations. This new notion of intersection information can successfully rule out the statistical relationships when responses are neither decoded by choice nor informative about stimulus. Moreover, intersection information is zero when responses affect the behavior but are not relevant to the stimulus. The paper tested the new measure intersection information with simulated data based on a scheme of perceptual discrimination task. By changing choice and sensory noise level, intersection information can reveal its desired property when the statistical relationship of S,R and C is different. Furthermore, the paper used intersection information to rank candidate neural codes for task performance with two experimental dataset. The paper appears to be technically sound. Properties of intersection information are clearly proved in the supplementary information. The problem and approach is not completely new, but it is a novel combination of familiar techniques. It's clear how this work differs from previous contributions. The related work is mostly referenced.   In summary, the intersection information criterion in this paper can offer single-trial quantification and capture more subtle statistical dependencies. It can help us to rank candidate neural codes and map information flow in brain. This is interesting work on a important question in neuroscience. But it would benefit from more thorough comparisons and analysis so that other people can be convinced to use this newly defined criterion.   Questions: Most importantly, the numerical calculation of intersection information is not fully discussed. This makes it hard to evaluate the practicality of the results. There should be more theoretical comparison between newly defined intersection information and shared information SI(C:{S;R}). It was shown in the section 2.1 that intersection information can rule out four scenarios of statistical relationship. The example in supplementary is very helpful and nicely designed. But I want to see the results of SI(C:{S;R}). It is not clear whether they are nonzero in scenario c and d. Moreover, in the framework of Partial Information Decomposition (PID) (Williams and Beer 2010), Interaction Information I(S;R;C) is defined to measure if the system has synergy or redundancy. This is equal to CI–SI in this paper. What is the advantage of the newly defined intersection information versus this Interaction Information? The test with simulated data showed some desired properties for intersection information. For example, when the choice noise increases, intersection information will decrease when intersection level is high, and the intersection information is smaller when sensory noise is larger. But why is there a crossing point under low sensory noise between high intersection scenario and low intersection scenarios, whereas under high sensory noise, there's no crossing point?

Reviewer 3



This paper addresses the issue of trying to determine whether neural activity recorded during perceptual discrimination is related to ultimate behavioral choice. The authors define a novel measure of intersection information that captures the amount of information between the presented stimulus S and the behavioral choice C that can be extracted from neural activity features R. They test their measure using simulated data and show that they recover the expected patterns with respect to choice and sensory noise and across two conditions of high and low intersection. Finally, they validate using experimental data from 2 perceptual discrimination tasks. It's generally well-written and does a good job of characterizing its various components regimes. I really liked the validation on real data and wish they would have explained these in more detail. The simulation was good to show as a proof of concept but I'm not sure I understood the main point, as there were too many conditions and parameters to keep track of (ie high/low intersection, choice noise, sensory noise). What are the relevant regimes for real data? Would you typically have a good estimate for sensory/choice noise and would thus know what regime of the model you're in? This was a dense paper and I'm not sure I understood it all (and it was a lot of content including the SI), but it seems like a useful, novel method of analyzing perceptual discrimination tasks. It'd be nice to see a discussion on whether this can be extended to other types of tasks as well, such as memory or RL tasks. A few minor errors: 1. Equation 1 has two identical terms; should one of these be UI(C: {R\S})? 2. Figure 4d legend should refer to panel c, not panel b 3. Line 209: "ensured that the timing variable did contain any rate information" should include a "not"?